# Mice Exposed to Combined Chronic Low-Dose Irradiation and Modeled Microgravity Develop Long-Term Neurological Sequelae

**DOI:** 10.3390/ijms20174094

**Published:** 2019-08-22

**Authors:** Eliah G. Overbey, Amber M. Paul, Willian A. da Silveira, Candice G.T. Tahimic, Sigrid S. Reinsch, Nathaniel Szewczyk, Seta Stanbouly, Charles Wang, Jonathan M. Galazka, Xiao Wen Mao

**Affiliations:** 1Department of Genome Sciences, University of Washington, Seattle, WA 98195, USA; 2Space Biosciences Division, NASA Ames Research Center, Moffett Field, CA 94035, USA; 3Universities Space Research Association, Columbia, MD 21046, USA; 4Institute for Global Food Security (IGF), School of Biological Sciences, Queen’s University, Belfast, Northern Ireland BT7 1NN, UK; 5KBR, Moffett Field, CA 94035, USA; 6MRC/ARUK Centre for Musculoskeletal Ageing Research & National Institute for Health Research Nottingham Biomedical Research Centre, Royal Derby Hospital, University of Nottingham, Derby DE22 3DT, UK; 7Division of Biomedical Engineering Sciences (BMES), Department of Basic Sciences, Loma Linda University, Loma Linda, CA 92354, USA; 8Center for Genomics, School of Medicine, Loma Linda University, Loma Linda, CA 92354, USA; 9Department of Basic Sciences, School of Medicine, Loma Linda University, Loma Linda, CA 92354, USA

**Keywords:** hindlimb unloading, chronic low-dose irradiation, brain, transcriptome

## Abstract

Spaceflight poses many challenges for humans. Ground-based analogs typically focus on single parameters of spaceflight and their associated acute effects. This study assesses the long-term transcriptional effects following single and combination spaceflight analog conditions using the mouse model: simulated microgravity via hindlimb unloading (HLU) and/or low-dose γ-ray irradiation (LDR) for 21 days, followed by 4 months of readaptation. Changes in gene expression and epigenetic modifications in brain samples during readaptation were analyzed by whole transcriptome shotgun sequencing (RNA-seq) and reduced representation bisulfite sequencing (RRBS). The results showed minimal gene expression and cytosine methylation alterations at 4 months readaptation within single treatment conditions of HLU or LDR. In contrast, following combined HLU+LDR, gene expression and promoter methylation analyses showed multiple altered pathways involved in neurogenesis and neuroplasticity, the regulation of neuropeptides, and cellular signaling. In brief, neurological readaptation following combined chronic LDR and HLU is a dynamic process that involves pathways that regulate neuronal function and structure and may lead to late onset neurological sequelae.

## 1. Introduction

The central nervous system (CNS) is vulnerable to irradiation [1] and fluid shifts [2] experienced during short- and long-term spaceflight. An important goal for the National Aeronautics and Space Administration (NASA) is to identify the effects of spaceflight-like conditions on the CNS to better prepare astronauts during long-duration missions to the Moon and Mars. During missions beyond low-earth orbit (LEO), astronauts will be continuously exposed to low-dose ionizing radiation (LDR). While high linear energy transfer (LET) galactic cosmic radiation (GCR) and low LET solar particle event (SPE) radiation contribute large portions of the radiation dose accumulated by astronaut crew members [3], interactions between GCR and SPE particles and spacecraft release secondary radiation including γ-rays that can deliver a significant fraction of the total mission does [4]. In addition, altered gravity (hypergravity experienced at launch/landing and microgravity experienced in-flight) affects homeostatic fluid-shifting in the CNS. Studies examining readaptation to Earth’s gravity are of equal importance, as recovery from spaceflight exposure may require medical intervention, in particular within organs known to be sensitive to irradiation and fluid shifts.

Astronauts in spaceflight experience impairments in neurocognition (as measured by altered decision making and problem solving) and neurobehavior (as measured by impaired visual perceptions, motor controls and sleep–wake cycles) [5]. The NASA Twins study offers some additional insight into post-flight cognitive decline that persists up to 6 months post-flight (end of experimental sampling) [6]. Longitudinal diffusion magnetic resonance imaging (MRI) data collected from astronauts pre- and post-flight (shuttle missions <30 days and ISS mission <200 days) revealed whole brain shifting within the skull and white matter disruptions, suggesting spaceflight leads to long-term alterations of brain structure in humans [7].

Rodent models offer an appropriate alternative to assess the effects of spaceflight-like conditions on brain tissue. Recently, the Rodent Research (RR)-1 mission flew mice to the International Space Station (ISS) for 33 days and measured neurobehavioral outcomes through video monitoring, which indicated 16-week old (at launch) females displayed unique group hyperactivity behavior at 2 weeks post-launch, implying neurological disruptions occur in flight [8]. Moreover, long-term readaptation to chronic LDR and hindlimb unloading (HLU) (a ground-based analog of spaceflight) resulted in increased aquaporin-4 (AQP4) protein expression, oxidative stress damage, apoptosis, blood–brain barrier (BBB) compromise and an increase in neurobehavioral risk-taking behavior [9,10,11].

The effects of acute radiation doses on the CNS have been well documented [12]. However, responses to chronic LDR exposure, as encountered on space missions (<0.04 Gy/y), are not well investigated [13], particularly in a pan-transcriptome context. Moreover, long-term readaptive outcomes on neurohealth following spaceflight-like conditions are less studied. Due to the need to better understand the transcriptional changes associated with chronic LDR in combination with reduced gravity (unloading) on neurohealth, mice were exposed to chronic LDR γ-ray irradiation (0.04 Gy), HLU, or a combination of HLU+LDR for 21 days, followed by a 4-month readaptation period. Whole transcriptome shotgun sequencing (RNA-seq) data were analyzed for differentially expressed genes (DEG), while data from reduced representation bisulfite sequencing (RRBS) were analyzed to determine differentially methylated promoters (DMP) [14]. Overall, we found minimal gene alterations at 4 months of readaptation within single treatment conditions of HLU or LDR, namely, in pathways related to the reduced expression of translational machinery, while the combination of HLU+LDR resulted in a wide panel of altered DEG and DMP profiles for tight junctions, aquaporins, neurogenesis markers and neuropeptides. Therefore, exposure to spaceflight-like conditions may lead to mild, long-term neurological consequences, while readaptation is an active process involving neuroplasticity and repair that is cooperatively engaged post-HLU+LDR exposure to maintain neural homeostasis.

## 2. Results

### 2.1. Experimental Design and Mouse Body Weights Across Experimental and Readaptation Timepoints

To assess the effects of prolonged spaceflight-like conditions on central nervous system (CNS) health following exposure, 6-month-old female *C57BL/6J* mice were either irradiated with low-dose γ-ray irradiation (0.04 Gy, LDR), hindlimb unloaded (HLU) or a combination of the two (HLU+LDR) for 21 days, followed by a 4-month readaptation period (Figure 1A). While the LDR γ-ray irradiation used does not recapitulate the entire space radiation spectrum, it constitutes a significant fraction and poses a hazard to astronauts’ health [15].

Total body weights were monitored from the initiation of the experimental conditions (0 days), throughout the duration of exposure (up to 21 days), and during the readaptation period (4 months, 4M). Mice were weighted weekly at the same time as the 21-day exposure and monthly for the 4-month readaptation phase. Baseline weight measurements were taken on day 0, prior to the initiation of HLU and LDR exposure. While there was no difference with LDR alone compared to control animals, a trend for decreased body weight in HLU and HLU+LDR combination conditions versus controls was noted, while statistical significance was observed at 7 days post-experimental start in HLU only (Figure 1B). Furthermore, during the readaptation period, decreased weights observed in HLU and in HLU+LDR animals recovered to control values, indicating physiological readaptation following exposure to spaceflight-like conditions.

Modeled microgravity in combination with chronic low-dose irradiation results in increased differential gene expression in brains at 4 months post-exposure. To determine the readaptation effects on the CNS at 4 months post-exposure, whole brains were isolated and mid-sagittal sectioned into two hemispheres. The right hemisphere was further sectioned mid-coronal and the caudal cortex (including hind- and midbrain) regions were collected for RNA-seq analysis. Collected brains were free of cerebellum and brainstem contamination. Principal component analysis (PCA) of RNA-seq data revealed clustering of samples by experimental conditions (Figure 2A), while volcano plots showed limited DEG (*p* < 0.05, Log_2_ fold change > 0.263) in all but the combination HLU+LDR (Figure 2B).

Brain readaptation following exposure to LDR alone displayed minimal (5) DEG with large fold change inductions only in genes of unknown function (Figure 3A, Table 1). Brain readaptation following exposure to HLU alone displayed 11 DEG with known functions primarily associated with reduced protein synthesis and/or functions mediated by ribosomes (Figure 3B, Table 1). In contrast to single exposures, brain readaptation following exposure to a combination of HLU+LDR displayed numerous (270) DEG associated with various cellular functions including tight junctions, aquaporins, neurogenesis and neuropeptide production (Figure 3C, Table 1). Comprehensive DEG lists are provided in the Appendix A. Collectively, these results showed minimal transcriptional changes in single treatment groups at 4 months post-exposure, while the combined exposure to HLU+LDR resulted in multiple DEG.

### 2.2. LDR, HLU and Combined HLU+LDR Exposures Displayed Minimal Gene and Promoter Methylation Overlap at 4 Months Readaptation 

Given that there were significant changes in gene expression in the combined exposure to HLU+LDR mice at the 4-month timepoint, we next determined whether these changes in gene expression reflected modifications to cytosine methylation using RRBS analysis. RRBS revealed 137, 71 and 170 differentially methylated promoters (DMPs) in the LDR, HLU, and HLU+LDR animals, respectively (Table 1). A full list of DMPs is provided in Appendix A. To find overlap across assays and experimental conditions, gene sets were created from DEG (upregulated/downregulated) and genes corresponding to DMPs (hypermethylated/hypomethylated). Overlapping genes between these sets were determined (Table 2) and summarized in an UpSet diagram [16] (Figure 4). Overall, there was little overlap between assays and experimental conditions (grey bars/dots/line). Most overlap was observed between DMPs from different experimental conditions (red bars/dots/lines). Overlap between DEGs from different experimental conditions was minimal (blue bars/dots/lines). Overlap between DEGs and DMPs from the same experimental condition or from different experimental conditions were also minimal (black bars/dots/lines). Collectively, these results indicated most DEGs and DMPs are unique to their experimental condition with minimal overlap between exposure groups.

### 2.3. Readaptation at 4 Months Following Exposure to Combined HLU+LDR Involves Gene Expression Changes in Multiple Brain-Associated Pathways

To determine the associated mechanisms involved in readaptation and neuroplasticity from simulated spaceflight combined HLU+LDR exposure, we used WebGestalt to detect enriched gene ontology (GO) terms in sets of up- and downregulated genes in post-HLU+LDR animals (Figure 5A,B, Appendix A). The downregulated set was enriched for GO terms associated with neurogenesis, transmembrane transport, cell signaling, and neurosynaptic and brain architecture deficits (Figure 5A). The upregulated DEG set was enriched for terms associated with cell proliferation, neurogenesis, and hormonal regulation (Figure 5B). Mouse Reactome pathway analyses of DEGs in the post-HLU+LDR animals indicated biological processes associated with signal transduction, neurodevelopment and neurohomeostasis (Appendix A). Human Reactome pathway translational analyses identified human orthologs involved in neurotransmission and cellular signaling (Appendix A and Table 3). Collectively, brain-readaptation post-combination HLU+LDR resulted in altered expression in pathways governing biological processes associated with neuroplasticity, neurogenesis, neurostructural reorganization and neuropeptide production.

## 3. Discussion

Herein, we identified that chronic exposure (21 days) to single or combined HLU and LDR resulted in long-term transcriptional alterations in CNS brain tissue in female mice at 4 months post-exposure. The gene expression changes observed here imply reduced transcriptional machinery, increased neurogenesis and neuropeptide production, and dysregulated cell structure and cell signaling. These studies offer translational value into pathways regulated in humans, as gene ontology analyses identified similar genetic overlap between mice and human profiles, in particular in pathways involved in neurodevelopmental homeostasis and signal transduction. Collectively, brain-related transcriptional changes that are dynamic in readaptation from exposure to spaceflight-like conditions may lead to long-term neurological consequences. Indeed, 21 days of exposure to combined HLU+LDR did not result in immediate neurobehavioral deficits (1 week post-exposure); however, increased risk-taking behavior was observed at 9 months post-exposure [9], indicating delayed neurological impairments in a combination of HLU+LDR, that was not apparent in HLU or LDR only.

To assess CNS effects, 4-month post-experimental conditions mid/hindbrain (sans cerebellum and brainstem) were collected, sectioned, and total RNA was isolated for RNA-seq analysis. This portion of the brain houses the hippocampus, basal ganglia, substantia nigra, and thalamus, and is a major signaling hub for the neuroendocrine system (hypothalamus and pituitary gland). Mid/hindbrain sections were assessed to localize the transcriptional profile to a region that is central to neurogenesis (hippocampus) and hypothalamic–pituitary axis (HPA) responses, in order to better assess neurofunctional output without potential contamination from the cerebral cortex or cerebellum. The results displayed robust DEG alterations within combination HLU+LDR at 4 months readaptation compared to controls, and marginal DEG alterations in single HLU or LDR at 4 months readaptation. At 4 months post-HLU exposure, DEGs including *EphrB3* and *Sh2d5* were altered, suggesting cell morphological and structural changes are active at this timepoint. At 4 months post-LDR exposure, only five DEGs were found, four of which have an unknown function. One DEG, *Rps13*, was reduced, suggesting impaired protein synthesis and/or other functional outputs mediated by ribosomes. Interestingly, *Rps13* was also reduced in HLU alone at 4 months readaptation along with other protein synthesis-related genes, including *Hist1h2bc* and *Rpl36a-ps2*, collectively suggesting fundamental translational pathways are altered during chronic HLU and LDR during readaptation. In line with this, another group has reported that chronic LDR whole brain irradiation resulted in protein synthesis impairment [1] that can affect adult neurogenesis [17]. Interestingly, no notable protein synthesis pathway deficits were identified in combined HLU+LDR exposures, suggesting distinct responses are generated within each experimental condition at readaptation. Moreover, very limited DEG overlap was observed between experimental conditions, further validating unique responses to each condition and highlighting the importance of implementing combined exposure studies to better assess spaceflight-like readaptation effects on the nervous system. Other spaceflight-like factors such as hypoxia and social isolation were not assessed in this study and merit further investigation. Nonetheless, the usefulness of this model provides a scaffold for future studies to assess the long-term effects of combination HLU+LDR on the health of the nervous system.

Brain structural damage through disruption of the blood–brain barrier (BBB) is associated with neurodegeneration and subsequent behavioral dysfunctions [18]. Our previous study showed that aquaporin-4 (AQP4), a water-channel protein involved in brain water homeostasis, was up-regulated following combined exposure to irradiation and unloading [9], suggesting a disturbance in BBB integrity may lead to edema [19]. Furthermore, AQP4 induction was seen in concert with subtle behavioral changes at 9 months post-exposure [9], indicating a potential role for BBB dysfunction and neurobehavioral consequences. Although *Aqp4* was not identified in this study, *Aqp1* gene expression was induced following combination of HLU+LDR at 4 months readaptation, suggesting similar BBB disturbances and possible shifts in brain water homeostasis [19]. In line with this, the induction of aquaporins may be linked to neuroinflammation [20,21,22], while the upregulation of additional immune activation markers including *Cd74*, *H2-Ab1, C1ql2* and *Ifit-3* were also identified in combination HLU+LDR at 4 months readaptation, along with hypomethylated *Cfi and Mx1*, collectively suggesting chronic neuroinflammation may persist during brain readaptation post-combined HLU+LDR exposure.

In response to extracellular cues (i.e., edema), cells rearrange their structures and ion transport proteins to maintain equilibrium [22,23,24]. Furthermore, neuroplasticity is a homeostatic repair mechanism that involves the formation of new synapses and neural processes to compensate for their communication loss following brain infarct, injury or disease [12]. These new formations result in brain restructuring and function, which may be occurring during readaptation post-combined HLU+LDR exposures. Indeed, our results fall in line with responses that were observed within brains isolated hours post-LDR (10 cGy) exposure, whereby multiple synaptic ion transport genes were altered [25]. Herein, multiple cell structural genes, i.e., *Drc7*, *Dnah7b* and *Fmnl1* and ion transport genes, i.e., *Gria3*, *Grin2a*, *Calb2*, *Scn5a*, *Clcn5*, *Kcnj13*, *Kcnn1* and *Kcnh1*, were altered post-combination HLU+LDR conditions. Moreover, hypomethylation analyses indicated altered cytoskeletal organization, i.e., *Ndel1*, *Cald1* and *Lmod3* were observed in both LDR and HLU, suggesting cellular structural changes involved in neuroplasticity and readaptation processes are active, in order to restructure and/or maintain brain homeostasis.

Tight junctions, including claudins, connexins, and occludins contribute to BBB permeability and maintenance. A reduction in these molecules has been implicated in acute BBB disruption in neuroviral [26] and neuroinflammatory diseases [27]. In line with this, *Cdh12* was downregulated, whereas, *Cldn2* was induced in combination of HLU+LDR. Furthermore, *Cdhr4* was hypomethylated in both LDR alone and HLU+LDR at 4 months readaptation, while *Cdhr2* was hypomethylated in LDR and HLU alone, but not in combination HLU+LDR at 4 months readaptation, suggesting BBB restructuring, and further indicating the dynamic process of brain physiology during readaptation.

Adult neurogenesis is a process of neural and glia self-renewal and the generation of progenitor cells from radial glial cells (RGCs) within the subventricular and subgranular zones of the hippocampus [28]. RGCs are quiescent and are activated to initiate neurogenesis in response to a stimulus [29], such as LDR [17,30] or models of increased intracranial pressure [31,32]. Indeed, the majority of highly upregulated DEGs post-combination HLU+LDR were involved in HPA signaling, neuropeptide growth factors, and neurogenesis, i.e., *Otp*, *Uncx*, *Ucn3*, *Avp*, *Pmch*, *Prox1*, *Calb2*, *Tbr-1*, and *Hcrt*. While the magnitude of gene expression does not always result in observable phenotypes, these notable DEGs may serve an important purpose in neuromaintenance during readaptation. Interestingly, *Crh*, a stress-induced hormone, was hypermethylated post-exposure to LDR and combination HLU+LDR at 4 months readaptation, implying transcriptional silencing of *Crh* may be engaged to suppress robust HPA activity during readaptation, which warrants further studies.

Readaptation results from combined exposures to HLU+LDR are in agreement with literature that spaceflight can alter neurofunction. Indeed, neurocognitive and behavioral effects via altered decision making, problem solving, visual perceptions, motor controls, and sleep–wake cycles have all been identified in flight [5]. While “space-brain” characteristics have suggested astronauts are at risk of developing neurological sequelae in flight [33] and post-flight [6,7]. Due to the limitations of human neurological studies, mouse models provide an excellent alternative, with the caveat that fluid shifts in mice are marginal in comparison to their human counterparts. Therefore, follow-up studies utilizing larger mammals, i.e., rats, may provide more robust responses. Further, readaptation effects from radiation have been described to cause differential nervous system responses in sex, age, type of irradiation, duration of exposure and total doses [1,12,34], while the readaptation effects of prolonged HLU on the neurohealth of different sexes and age groups are still unclear. Therefore, future studies approaching these additional demographics, both modeled in mice or rats, are required. Nonetheless, our readaptation study in mice exposed to combined HLU+LDR displayed a panel of neurological changes that were translational to humans, suggesting this is a suitable animal model to study spaceflight-related readaptation effects on the nervous system. Additionally, future studies should expand the types of radiation exposure animals experience in order to fully capture the spectrum of radiation that bombards astronauts during spaceflight. The space radiation environment beyond LEO contains several types of ionizing radiation. Galactic cosmic radiation (GCR) with relatively high linear energy transfer (LET) and proton radiation due to solar particle events (SPEs) contribute a significant portion of the radiation dose accumulated by astronaut crew members [3]. High-energy heavy ions (HZE radiation) can produce distinct patterns of energy deposition in cells and tissues [13,35]. However, secondary particles produced by the interaction of SPE protons and heavy-charged GCR particles with the spacecraft structure include γ-rays and x-rays [3]. These radiation sources can also deliver a significant fraction of the total mission dose and pose a hazard to astronauts’ health [15]. For chronic irradiation exposure, we used flood sources consisting of plastic sheets embedded with the isotope ^57^Co that emits medium energy (122 and 136 keV) γ-rays. Additional research is needed to elucidate the CNS damage and readaptation response induced by the full spectrum of space radiation.

In summary, a multitude of neurological effects and persistent brain transcriptional alterations were observed at 4 months post-HLU+LDR in 21-day exposures. Changes included reduced transcriptional machinery, increased neurogenesis and neuropeptide production, and dysregulated cell structure and signaling genes. Thus, brain-related transcriptional changes are dynamic and plastic during readaptation from exposure to spaceflight-like conditions, and while these exposures may lead to long-term neurological consequences, the active processes of neuroplasticity and repair are engaged to maintain neural homeostasis.

## 4. Materials and Methods

### 4.1. Animals

Six-month-old, female *C57BL/6J* mice (Jackson Laboratory), as previously reported [10], were utilized in this report. Upon arrival, they were acclimatized for 7 days in standard habitats at 20 °C with a 12 h:12 h light:dark cycle. Commercial pellet chow and hydrogel were available ad libitum. Animal health status, water and food intake were monitored daily. The study followed recommendations in the Guide for the Care and Use of Laboratory Animals [36] and was approved on April 30, 2014, by the Institutional Animal Care and Use Committee (IACUC) at Loma Linda University (Protocol number 8130028).

### 4.2. Experimental Conditions

Following acclimatization, animals were housed one per cage and assigned to one of four groups: (1) control (*n* = 3); (2) hindlimb unloaded (HLU) (*n* = 6); (3) low-dose irradiated (LDR) (*n* = 6); and (4) combination hindlimb unloaded and low-dose irradiated (HLU+LDR) (*n* = 4) for 21 days. Post-experimental exposure animals were group housed (3 per cage) for up to 4 months (4M).

### 4.3. Hindlimb Unloading and Whole-Body Irradiation

Hindlimb-unloaded animals (HLU) were treated as previously described [37], with animals maintained at a 35–40 head-down tilt with the hindlimbs elevated above the bottom of the cage. LDR gamma irradiation was performed by placing ^57^Co plates (185 MBq activity; GPI, Stoughton, WI) under the cages (1 plate per 2 cages). These cobalt plates deliver chronic low-dose/low-dose-rate γ-ray irradiation. A total dose of 0.04 Gy was delivered at a rate of 0.01 cGy/h over 21 days. HLU+LDR animals were both unloaded and irradiated as described here. Control animals received no treatment. All animals were handled similarly.

### 4.4. Euthanasia, Dissection, Tissue Storage, and Nucleic Acid Extraction

Mice were euthanized with 100% CO_2_ followed by immediate exsanguination by cardiac puncture. Immediately after euthanasia brains were removed and bisected along the midline and coronally within the half hemispheres, with the brainstem and cerebellum removed. The right caudal half hemisphere of the brain (containing mid- and hindbrain) from each mouse was placed in a sterile cryovial, snap frozen in liquid nitrogen and kept at −80 °C. AllPrep DNA/RNA/miRNA Universal Kit (Qiagen, Hilden, Germany) was used to extract RNA and DNA according to the manufacturer’s instructions. Tissue homogenization was performed using the Tissue homogenizing CKMix (Bertin Instruments, Montigny-le-Bretonneux, France) on a Minilys homogenizer (Bertin Instruments). RNA and DNA concentrations were measured using a Qubit 3.0 Fluorometer (Thermo Fisher Scientific, Waltham, MA, USA) and stored at −80 °C.

### 4.5. RNA Sequencing

The Ovation Mouse RNA-Seq System 1–96 (NuGEN Technologies, Redwood City, CA, USA) was used per manufacturer’s instructions to construct RNA-Seq libraries. A total of 100 ng of total RNA was used as input. First and second strands of cDNA were synthesized from total RNA spiked with 1 µL of 1:500 diluted ERCC ExFold RNA Spike-In Mix 1 (Life Technologies, Carlsbad, CA, USA) at the appropriate ratio. Following primer annealing and cDNA synthesis, the products were sheared using Covaris S220 Focused-ultrasonicator (Covaris Inc., Woburn, MA, USA) to obtain fragment sizes between 150–200 bp. This was followed by end-repair, adaptor index ligation and strand selection. Barcodes with unique indices out of 96 indices, were used per sample for multiplexing. Strand selection was performed using a custom InDA-C primer mixture SS5 Version5 for mice (NuGEN Technologies). Libraries were amplified by PCR with 17 cycles on a Mastercycler Pro (Eppendorf) and purified with RNAClean XP Agencourt beads (Beckman Coulter, Pasadena, CA, USA). These libraries were sequenced on a HiSeq 4000 (Illumina, Mira Loma, CA, USA) to generate 15–30 M 75-bp single end reads per sample. Raw data are available at NASA GeneLab (genelab.nasa.gov, accession GLDS-202) [14].

### 4.6. Reduced Representation Bisulfite Sequencing (RRBS)

The RRBS library was constructed with 100 ng of gDNA as input using Ovation RRBS Methyl-Seq System (NuGEN Technologies) according to the manufacturer’s protocol. Briefly, MspI enzyme, which cuts the DNA at CCGG sites, was used to digest gDNA into fragments. The fragments were directly subject to end blunting and phosphorylation followed by ligation to a methylated adapter with a single-base T overhang. The ligation products were finally repaired in a thermal cycler under the program (60 °C for 10 min, 70 °C for 10 min, held at 4 °C). The product of the final repair reaction was used for bisulfite conversion using the EpiTect Fast DNA Bisulfite Kit (Qiagen). Bisulfite-converted DNA was then amplified on a Mastercycler Pro (Eppendorf, Hamburg, Germany) and bead-purified with Agencourt RNAClean XP Beads (Beckman Coulter). The RRBS libraries were sequenced on Illumina HiSeq 4000 at Loma Linda Center for Genomics to generate 15–55 M 70-bp single end reads per sample. Raw data are available at NASA GeneLab (genelab.nasa.gov, accession GLDS-202) [14].

### 4.7. Differential Expression and Pathway Enrichment Analysis

Sequencing reads were first trimmed with Cutadapt [38] using the Trim Galore! wrapper. Reads were mapped with STAR [39], expression quantified with RSEM [40] and tximport [41] was used to import counts into DESeq2 [42] for the detection of differentially expressed genes. Log_2_ fold-change and adjusted *p*-value cutoffs of 0.263 and 0.05 were used, respectively. Enriched GO terms were detected with WebGestalt [43] using the biological process database. Enriched pathways were detected with Reactome version 67 [44] using an overrepresentation analysis and the full list of DEGs. Human pathways were found using Reactome’s “Project to human” option. Heatmaps were generated using the webtool Morpheus, from the Broad Institute (https://software.broadinstitute.org/morpheus/) by performing hierarchical clustering using Euclidean distance. The volcano plots in Figure 2 were generated with the R package EnhancedVolcano. The UpSet diagram in Figure 4 was generated using Intervene [16].

### 4.8. Differential Methylation Analysis

Sequencing reads were first trimmed with Cutadapt [38] using the Trim Galore! wrapper using the “--RRBS” flag before processing with the “bismark” and “bismark_methylation_extractor” scripts from Bismark [45]. The differential methylation of gene bodies and promoters was performed with RnBeads version 1.10.8 and R version 3.4.1 (R Foundation, Boston, MA, USA) [46]. The combined ranking score was utilized to detect differential methylation. It combines absolute and relative effect sizes as well as *p*-values from statistical modeling into a single score.

## Figures and Tables

**Figure 1 ijms-20-04094-f001:**
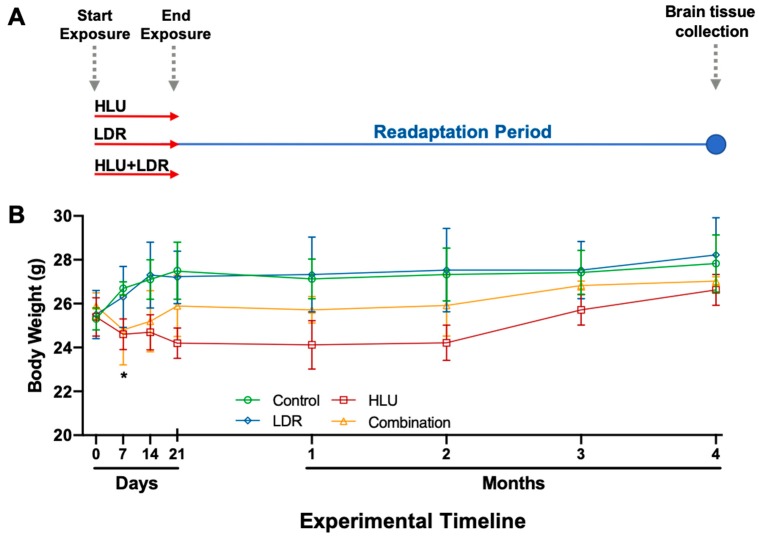
Experimental timeline and body weights during- and post-hindlimb unloading (HLU), low-dose irradiation (LDR) or combined HLU+LDR. (**A**) Whole brains were isolated from C57BL/6J at 4 months post-experimental conditions of hindlimb unloading (HLU, 21 days), low-dose irradiation (LDR, 0.04 Gy for 21 days) or combined LDR and HLU timeline. (**B**) Body weights were monitored throughout the experiment (21 days) and during readaptation (4 months post-experimental exposure). Baseline weights were taken on day 0, prior to the initiation of HLU and LDR exposure. An unpaired *t*-test with Welch’s correction compared experimental groups with controls. A * denotes significance of HLU only compared to controls at the 7-day timepoint. Error bars denote standard error of means. Dashed gray arrows highlight start and endpoints of experimental LDR and HLU treatments, and the final timepoint of the experiment when tissue was collected for transcriptomic analysis.

**Figure 2 ijms-20-04094-f002:**
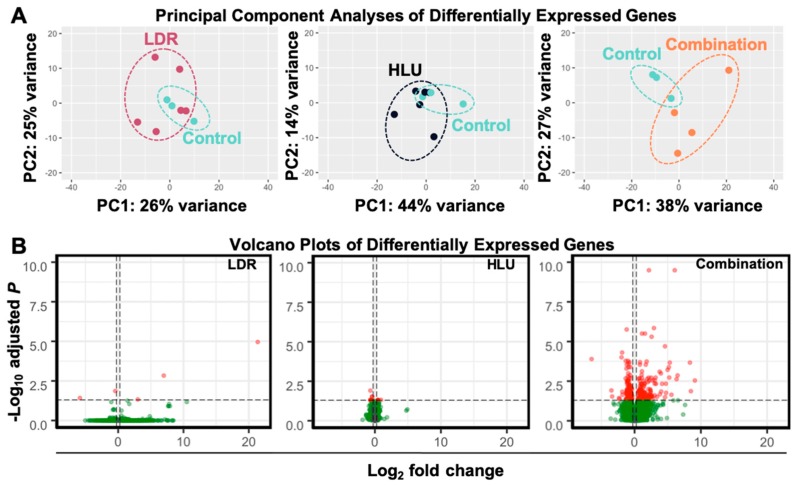
HLU in combination with LDR results in a heightened differentially expressed gene (DEG) profile in brains isolated at 4 months post-experimental conditions. DEG identified in brain isolated at 4 months post-LDR (0.04 Gy for 21 days), HLU (21 days) or combined HLU+LDR. (**A**) Principal component analysis (PCA) displays sample clustering within each experimental condition (*n* = 3–6). (**B**) Volcano plots indicate DEG profiles at 4 months post-experimental conditions. Vertical dashed lines indicate the Log_2_ fold-change cutoff used (0.263), while horizontal dashed lines indicate the adjusted *p*-value cutoff used (0.05). Red dots indicate differentially expressed genes that meet our adjusted *p*-value and Log_2_ fold-change cutoffs and compose our set of differentially expressed genes. Green dots indicate genes that do not meet our adjusted *p*-value cutoff and, therefore, are not differentially expressed. Darker saturation of color represents overlap of genes.

**Figure 3 ijms-20-04094-f003:**
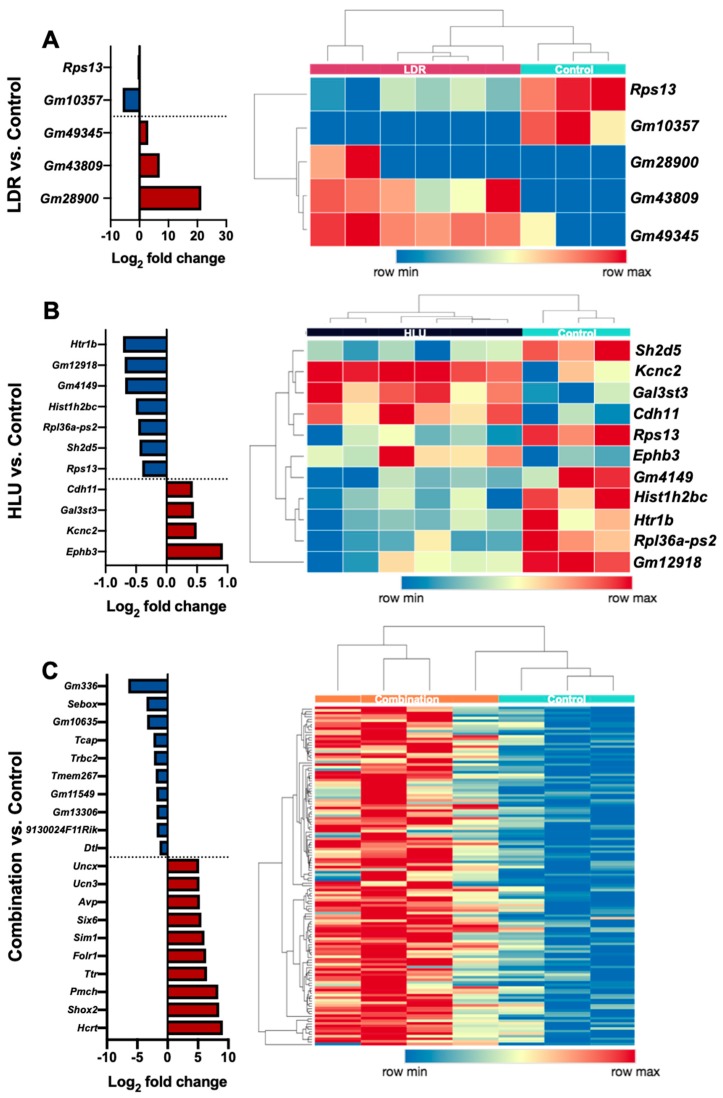
DEG profiles of brains isolated at 4 months post-HLU, LDR or combined HLU+LDR. (**A**, left panel) Graph depicts LDR versus control samples Log_2_ fold-change mean values (*n* = 3–6); (**A**, right panel) Heatmap displays individual DEG. (**B**, left panel) Graph depicts HLU versus control samples Log_2_ fold-change mean values (*n* = 3–6); (**B**, right panel) Heatmap displays individual DEG. (**C**, left panel) Graph depicts a subset of the largest Log_2_ fold-change gene set within combined HLU+LDR (combination) verses control samples Log_2_ fold-change mean values (*n* = 3–4); (**C**, right panel) Heatmap displays individual DEG.

**Figure 4 ijms-20-04094-f004:**
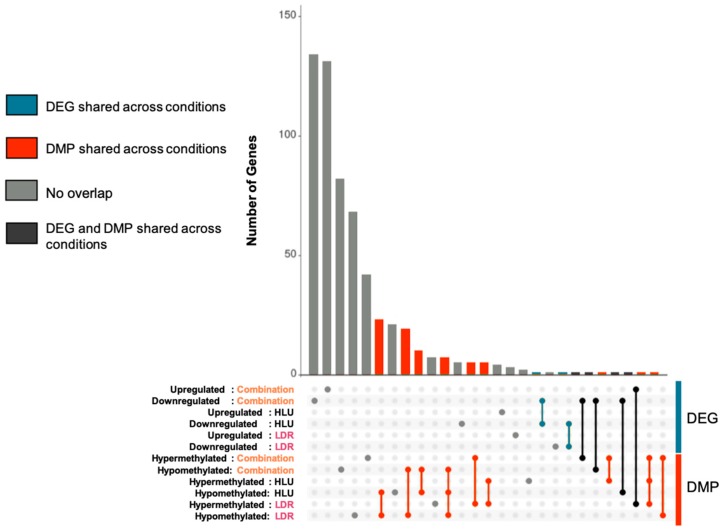
Minimal brain differential gene expression and promoter methylation overlap across different simulated spaceflight experimental conditions. Gene lists of DEGs and the genes corresponding to the differentially methylated promoters (DMPs) were created for each condition: LDR, HLU, or HLU+LDR combined versus controls. The dotted region depicts all conditions, and the connecting lines depict DEG/DMP overlap between experimental conditions. The vertical bars indicate the size of that overlap. A single, grey dot indicates the number of DEG/DMP that do not overlap with any other DEG/DMP. The red dots indicate DMP overlap across different conditions. The blue dots indicate DEG overlap across different conditions. The black dots indicate DEG and DMP overlap. “Upregulated/downregulated” denotes gene expression of DEG and “hypermethylated/hypomethylated” denotes methylation patterns of DMPs.

**Figure 5 ijms-20-04094-f005:**
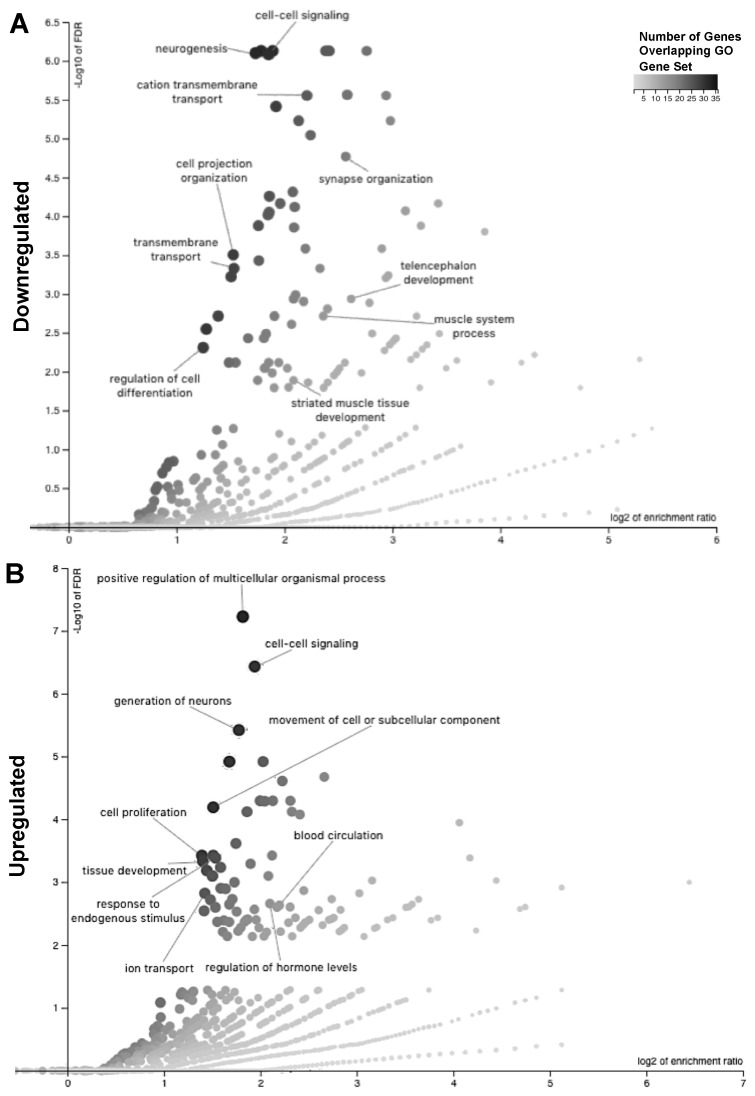
Altered biological processes and pathways identified at 4 months post-combined HLU+LDR exposure. (**A**,**B**) WebGestalt graphs depict enriched gene ontology (GO) terms for downregulated (**A**) and upregulated (**B**) DEGs. The x-axis measures the enrichment ratio for each GO term on a Log_2_ scale. The y-axis measures -Log_10_ (false discovery rate, FDR). The size and color tone of the dot is proportional to the size of the category. The displayed categories are from WebGestalt’s “Weighted set cover” option, which displays the GO categories that minimize redundancy.

**Table 1 ijms-20-04094-t001:** Differentially methylated promoters (DMPs) and differentially expressed genes (DEGs) at 4 months post-exposure conditions.

	DMP			DEG		
	Total DMP	Hypomethylated DMP	Hypermethylated DMP	TotalDEG	Upregulated DEG	Downregulated DEG
**Radiation vs. Control**	*137 promoters*	*118 promoters*	*19 promoters*	*Gm28900*	*Gm28900*	*Rps13*
				*Gm43809*	*Gm43809*	*Gm10357*
				*Gm49345*	*Gm49345*	
				*Rps13*		
				*Gm10357*		
**Unloading vs. Control**	*71 promoters*	*62 promoters*	*Gm35801*	*Ephb3*	*Ephb3*	*Gm12918*
			*Gm13066*	*Kcnc2*	*Kcnc2*	*Gm4149*
			*Cnpy1*	*Cdh11*	*Cdh11*	*Hist1h2bc*
			*Cbfa2t2-ps1*	*Gal3st3*	*Gal3st3*	*Htr1b*
			*Gm15700*	*Gm12918*		*Sh2d5*
			*Gm27679*	*Gm4149*		*Rps13*
			*Hbq1a*	*Hist1h2bc*		*Rpl36a-ps2*
			*1700019N19Rik*	*Htr1b*		
			*Gm6024*	*Sh2d5*		
				*Rps13*		
				*Rpl36a-ps2*		
**Combined vs. Control**	*170 promoters*	*119 promoters*	*51 promoters*	*270 genes*	*132 genes*	*138 genes*

**Table 2 ijms-20-04094-t002:** Gene list of overlapping differentially expressed genes (DEGs) and differentially methylated promoters (DMPs) within each post-exposure condition.

**DEG/DEG Overlap**	
**Condition**	**Gene**
Down HLU+LDR/Down HLU	*Gm12918*
Down HLU/Down LDR	*Rps13*
**DMP/DMP Overlap**	
**Condition**	**Methylated promoter of gene**
Hypo LDR/Hypo HLU	*Gm38049; Zfp648; 1700018C11Rik;*
	*B230104I21Rik; Cald1; Lmod3*
	*Smco3; Angpt2; Gm28515*
	*Sln; Gm27159; Gm15271*
	*4933405E24Rik; Ndel1; Gm11373*
	*Gm5083; Cdhr2; Mir8097*
	*1700011A15Rik; Krt8; Mir8094*
	*Pcdh12; Tmem180*
Hypo LDR/Hypo HLU+LDR	*Gm16564; Gm37267; Adamtsl2*
	*Gm13413; Bbox1; Gm11805*
	*1700071G01Rik; Prr33; Cdhr4*
	*Slfn4; Krt19; Gm6401*
	*Gm15516; Mx2; Gm10268*
	*Gm25301; Armcx2; Gm15726*
	*Gm15247*
Hypo HLU/Hypo HLU+LDR	*Gm29100; Nav1; Gm29282*
	*Cfi; Mypopos; Adgb*
	*Gm26535; Abi3; Gm6969*
	*Lpar6*
Hyper LDR/Hyper HLU+LDR	*Otos; Sys1; Klhl34*
	*Crh; Pcdhb12*
Hyper HLU/Hyper HLU+LDR	*Gm35801*
Hypo LDR/Hyper HLU+LDR	*Gm6987*
Hyper HLU/Hyper LDR	*1700019N19Rik; Cnpy1; Gm6024*
	*Gm15700; Gm27679*
Hypo LDR/Hypo HLU/Hypo HLU+LDR	*Trp53inp2; Gm22518; Dio1*
	*Mir704; Gm805; Gm24998*
	*Gm15648*
Hyper LDR/Hyper HLU/Hyper HLU+LDR	*Cbfa2t2-ps1*
**DEG/DMP Overlap**	
**Condition**	**Gene or Methylated promoter of gene**
Down HLU+LDR/Hyper HLU+LDR	*Gm7120*
Down HLU+LDR/Hypo HLU+LDR	*Ephx4*
Down HLU+LDR/Hypo HLU	*Blnk*
Up HLU+LDR/Hyper LDR	*Rgag4*

**Table 3 ijms-20-04094-t003:** Combination of LDR and HLU versus controls DEG and Reactome pathway IDs in mice that translated to human orthologs.

Pathway Name	Associated Mouse Genes	Mouse Pathway ID	Human Pathway ID
Neuronal System	*Gria3, Cacnb3, Hcn1, Camkk2, Kcnh1, Kcnma1, Kcnn1, Stx1a, Kcnv1, Kcnh5, Grin2a, Slc17a7, Dlgap1, Homer1, Snap25, Nrgn, Kcnh7, Slitrk1*	R-MMU-112316	R-HSA-112316
Potassium Channels	*Kcnn1, Kcnh5, Kcnma1, Hcn1, Kcnh1, Kcnv1, Kcnh7*	R-MMU-1296071	R-HSA-1296071
Voltage Gated Potassium Channels	*Kcnv1, Kcnh1, Kcnh5, Kcnh7*	R-MMU-1296072	R-HSA-1296072
cGMP effects	*Pde1a*	R-MMU-418457	R-HSA-418457
p75 NTR receptor-mediated signaling	*Rtn4r, Obscn, Lingo1, Kalrn*	R-MMU-193704	R-HSA-193704
Nitric oxide stimulates guanylate cyclase	*Pde1a*	R-MMU-392154	R-HSA-392154

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
