# Peer review of "Mice Exposed to Combined Chronic Low-Dose Irradiation and Modeled Microgravity Develop Long-Term Neurological Sequelae"

_ijms, 2019, doi:10.3390/ijms20174094_

Round 1

Reviewer 1 Report

The manuscript by Overbey and colleagues details transcriptional changes following 4 months of readaptation after 21 days of exposure to hindlimb unloading (HLU), low dose gamma radiation exposure (LDR), the combination (HLU+LDR), and an untreated sham group (control). The authors report many interesting changes following these manipulations and what appear to be novel changes induced by readaptation to the combination of HLU + LDR that are not evident at 4 months following the exposures to the single variables. The data are especially interesting because they are examining the effects of multiple spaceflight factors on the CNS, which is a relatively novel area of investigation. My major and minor comments appear below. 

The authors need to more fully justify their use of gamma rays and not HZE irradiation for this study. This reviewer understands the difficulties with a 21-day chronic exposure to HZE radiation, however, the authors do not adequately address this issue and simply state (line 337), "This model despite inherent limitations, takes into account extended exposure to low dose ionizing radiation which astronauts will encounter in long duration spaceflight missions." This is too simplistic of an explanation. This is only exposure to the low LET component of space radiation and there is no exposure to the high LET component of space radiation, nor any mention of the high LET component compared to the low LET component. This fact needs to be divulged earlier in the manuscript and it needs to be justified why this model is relevant to long duration spaceflight. The authors briefly mention the fact that gamma rays can be produced by  interactions of particles with shielding material, but more details are needed. Authors should also cite Walker, Townsend, and Norbury, 2013, Advances in Space Research, 51, 1792-1799.

The authors switch between stating these effects are the result of chronic HLU and 4 months of readaptation following chronic HLU. Since no measurements were taken immediately at the end of the HLU manipulation and all measurements were acquired after 4 months of loading, the authors need to consistently address these changes as changes related to chronic HLU followed by 4 months of readaptation. For example, line 234, the authors argue that this data supports the hypothesis that many pathways are altered during chronic HLU, however, this data is actually showing the pathways that are altered following 4 months of readaptation following 21 days of HLU. Without measurements taken immediately following the conclusion of HLU, it is not possible to separate the effects of HLU alone and HLU + 4 months of readaptation in the current study. 

The authors make claims that weight loss could have contributed to the changes found in the HLU groups. However, this weight loss is not statistically significant (except for one time point) and is arguably not biologically significant, given that is appears to be no more than a loss of 1-2 grams from the HLU groups' baseline weights (~3-5% change). Compared to the control group, it is only about a 2-3 gram loss over many days (i.e., animals are losing weight very slowly). This hardly qualifies as "significant" weight loss. This is even more apparent because this weight loss is measured weekly and then monthly, not daily. Further, the citations used are not supporting this type of weight loss in mice as being significant and affecting brain function, which are needed for this argument. Without some other data (e.g., total food consumed, total ml water consumed, transcriptional changes at these time points when weights were measured, behavioral changes) to support the assumption that these animals were experiencing "significant" weight loss, this argument should be removed from the manuscript. It distracts from and weakens the current results of the paper that these changes are following the readaptation from HLU+LDR and re unique compared to each variable alone. 

The authors need to address how group housing could have influenced the results, since this is how the animals were housed during the 4-month readaptation period, but not how they were housed during the 21-day spaceflight factor exposure period. Further, more details are needed for how the groups of mice were caged - e.g., were all HLU animals housed together (2 cages of 3 mice) or were the groups randomly split between cages? 

Minor comments:

Line 100 - states weights were monitored "periodically" - please state exactly when weights were monitored, i.e., weekly for the first 21 days and monthly for the 4 month readaptation phase. Please include whether this was at the same time each day for each group of animals. Also include when the baseline weight measurement was taken (e.g., day immediately prior to unloading or IR exposure beginning?). 

Line 104 - change "display" to induce or cause

Figure 2 - volcano plots - all dots appear grey on the review version of the manuscript - dark grey or light grey. Is the darker grey color supposed to be black? If so, this color needs to be changed to black. Or are these darker patches areas where many light grey dots overlap?  If so, this needs to be stated. Regardless, the caption needs to more clearly define these dots and their colors. 

Line 155 - remove indent since this is a continuation of the sentence from the previous page. 

Line 160 - remove extra space between "following" and "exposure"

Line 176 - no reference to Table 3 in text. Please include this. 

Line 217 - Change to HLU-reduced

Author Response

The manuscript by Overbey and colleagues details transcriptional changes following 4 months of readaptation after 21 days of exposure to hindlimb unloading (HLU), low dose gamma radiation exposure (LDR), the combination (HLU+LDR), and an untreated sham group (control). The authors report many interesting changes following these manipulations and what appear to be novel changes induced by readaptation to the combination of HLU + LDR that are not evident at 4 months following the exposures to the single variables. The data are especially interesting because they are examining the effects of multiple spaceflight factors on the CNS, which is a relatively novel area of investigation. My major and minor comments appear below. 

Response: The authors thank the reviewer for the positive comments and acknowledgement of the importance of the study. 

The authors need to more fully justify their use of gamma rays and not HZE irradiation for this study. This reviewer understands the difficulties with a 21-day chronic exposure to HZE radiation, however, the authors do not adequately address this issue and simply state (line 337), "This model despite inherent limitations, takes into account extended exposure to low dose ionizing radiation which astronauts will encounter in long duration spaceflight missions." This is too simplistic of an explanation. This is only exposure to the low LET component of space radiation and there is no exposure to the high LET component of space radiation, nor any mention of the high LET component compared to the low LET component. This fact needs to be divulged earlier in the manuscript and it needs to be justified why this model is relevant to long duration spaceflight. The authors briefly mention the fact that gamma rays can be produced by  interactions of particles with shielding material, but more details are needed. Authors should also cite Walker, Townsend, and Norbury, 2013, Advances in Space Research, 51, 1792-1799.

Response: We understand heavy ions produce distinct patterns of energy deposition in cells and tissues compared to gamma-rays leading to large uncertainties in risk estimates [1]. However, the damage induced by space radiation-relevant low-dose low LET radiation (gamma rays and x-ray) are still important to document and investigate [2]. Cobalt plates are a feasible model to deliver chronic low-dose/low-dose-rate irradiation while animals are unloaded. Given the scarcity of data in the field related to prolong exposure of irradiation and simulated microgravity on the CNS, we believe our findings are relevant and important for astronaut health risk assessment and for improving our knowledge of spaceflight effects on nervous tissue and provide a basis for further ground-based or spaceflight experiments to study the mechanism of spaceflight-induced molecular changes described in our study.

We agree that it is important to clarify relevancy of gamma-ray irradiation in context of the entire spectrum of space radiation, and have added brief statements in the introduction and results sections and a longer discussion of the issue in the discussion section. We have added a citation to Walker et al [3].

The authors switch between stating these effects are the result of chronic HLU and 4 months of readaptation following chronic HLU. Since no measurements were taken immediately at the end of the HLU manipulation and all measurements were acquired after 4 months of loading, the authors need to consistently address these changes as changes related to chronic HLU followed by 4 months of readaptation. For example, line 234, the authors argue that this data supports the hypothesis that many pathways are altered during chronic HLU, however, this data is actually showing the pathways that are altered following 4 months of readaptation following 21 days of HLU. Without measurements taken immediately following the conclusion of HLU, it is not possible to separate the effects of HLU alone and HLU + 4 months of readaptation in the current study. 

Response: We thank the Reviewer for this comment and have made the necessary changes throughout the manuscript.

The authors make claims that weight loss could have contributed to the changes found in the HLU groups. However, this weight loss is not statistically significant (except for one time point) and is arguably not biologically significant, given that is appears to be no more than a loss of 1-2 grams from the HLU groups' baseline weights (~3-5% change). Compared to the control group, it is only about a 2-3 gram loss over many days (i.e., animals are losing weight very slowly). This hardly qualifies as "significant" weight loss. This is even more apparent because this weight loss is measured weekly and then monthly, not daily. Further, the citations used are not supporting this type of weight loss in mice as being significant and affecting brain function, which are needed for this argument. Without some other data (e.g., total food consumed, total ml water consumed, transcriptional changes at these time points when weights were measured, behavioral changes) to support the assumption that these animals were experiencing "significant" weight loss, this argument should be removed from the manuscript. It distracts from and weakens the current results of the paper that these changes are following the readaptation from HLU+LDR and re unique compared to each variable alone. 

Response: We thank the Reviewer for this important insight and have removed this argument from the manuscript.

The authors need to address how group housing could have influenced the results, since this is how the animals were housed during the 4-month readaptation period, but not how they were housed during the 21-day spaceflight factor exposure period. Further, more details are needed for how the groups of mice were caged - e.g., were all HLU animals housed together (2 cages of 3 mice) or were the groups randomly split between cages? 

Response:  Animals were group-housed by their exposure group during the 4-month readaptation period, e.g. LDR mice were housed in the same cage. Since mice are social animals, whenever possible we group-housed them to avoid possible non-specific, stress-induced effects. Further, female mice were used to reduce potential physical aggression (wound/injury) that has been observed between group-housed male mice. Therefore, group-housing and female mice would minimize the potential for stress-related changes in physiology that could affect research outcomes. Furthermore, these studies support rodent studies on the ISS, whereby mice are group-housed, therefore this research can provide a suitable comparison to  flight studies.

Minor comments:

Line 100 - states weights were monitored "periodically" - please state exactly when weights were monitored, i.e., weekly for the first 21 days and monthly for the 4 month readaptation phase. Please include whether this was at the same time each day for each group of animals. Also include when the baseline weight measurement was taken (e.g., day immediately prior to unloading or IR exposure beginning?). 

Response: Mice were weighed weekly at the same time for the first 21 days and monthly for the 4-month readaptation phase. The baseline weight measurement were performed on day 0, prior to initiation of HLU or LDR. The manuscript has been updated with this information (Page 3, Line 102-104) and within the Figure 1 legend.

Line 104 - change "display" to induce or cause

Response: “display” has been changed to “cause”.

Figure 2 - volcano plots - all dots appear grey on the review version of the manuscript - dark grey or light grey. Is the darker grey color supposed to be black? If so, this color needs to be changed to black. Or are these darker patches areas where many light grey dots overlap?  If so, this needs to be stated. Regardless, the caption needs to more clearly define these dots and their colors. 

Response: The dark/light grey has been changed to red/green for clarity and a description of these have been included in the figure legends (Page 4, Line 126-128).

Line 155 - remove indent since this is a continuation of the sentence from the previous page. 

Response: The indentation was removed.

Line 160 - remove extra space between "following" and "exposure"

Response: The extra space was removed.

Line 176 - no reference to Table 3 in text. Please include this. 

Response: Table 3 is cited in the text next to Fig. S1B.

Line 217 - Change to HLU-reduced

Response: This change has been made.

References

Cucinotta, F.A.; Cacao, E. Risks of Cognitive Detriments after Low Dose Heavy Ion and Proton Exposures. Int. J. Radiat. Biol. 2019, 1–36. Chatterjee, S.; Pietrofesa, R.A.; Park, K.; Tao, J.-Q.; Carabe-Fernandez, A.; Berman, A.T.; Koumenis, C.; Sielecki, T.; Christofidou-Solomidou, M. LGM2605 Reduces Space Radiation-Induced NLRP3 Inflammasome Activation and Damage in In Vitro Lung Vascular Networks. Int. J. Mol. Sci. 2019, 20. Walker, S.A.; Townsend, L.W.; Norbury, J.W. Heavy ion contributions to organ dose equivalent for the 1977 galactic cosmic ray spectrum. Adv. Space Res. 2013, 51, 1792–1799.

Reviewer 2 Report

The study has a new insght about LDR+HLU model.

The presentation should be a little more improved.

The authors should show how they determine the protocol of this study.

Figure 2 needs ftitle.

Author Response

The study has a new insight about LDR+HLU model.

Response: The authors thank the reviewer for the acknowledging the novelty of the study.

The presentation should be a little more improved.

Response: We thank the Reviewer this comment. Although we believe the organization of the manuscript is logically arranged, we have provided revisions to certain sections of the manuscript that should improve clarity.

The authors should show how they determine the protocol of this study.

Response: Long-term impact of chronic exposure of low dose/low-dose-rate radiation on physiological system in combination with simulated microgravity, in particular within the nervous system, are not well documented. Very few studies have been performed with space mission-relevant irradiation doses (<0.04 Gy/y) combined with prolonged simulated microgravity in rodent models [1]. Therefore, in order to better simulate spaceflight conditions, a mouse model of combined exposure to radiation 0.04Gy and unloading for 21 days (comparable timeframe to spaceflight rodent missions) was used in our study. Furthermore, ground-based readaptation effects on the CNS from spaceflight exposure are also not well-studied. Therefore, this study offers insight into the readapative process of the nervous system as a result of spaceflight-relevant simulated microgravity in combination with irradiation.

Figure 2 needs title.

Response: We have included titles in Figure 2 for clarity

References:

1. Cucinotta, F.A.; Cacao, E. Risks of Cognitive Detriments after Low Dose Heavy Ion and Proton Exposures. Int. J. Radiat. Biol. 2019, 1–36.